# Antimicrobial Effects and Antioxidant Activity of *Myrtus communis* L. Essential Oil in Beef Stored under Different Packaging Conditions

**DOI:** 10.3390/foods12183390

**Published:** 2023-09-10

**Authors:** Dirce Moura, Joana Vilela, Sónia Saraiva, Filipe Monteiro-Silva, José M. M. M. De Almeida, Cristina Saraiva

**Affiliations:** 1Animal and Veterinary Science Center (CECAV), University of Trás-os-Montes e Alto Douro (UTAD), Quinta de Prados, 5000-801 Vila Real, Portugal; didi.dirce93@hotmail.com (D.M.); joana.vilela@live.com.pt (J.V.); soniasaraiva@utad.pt (S.S.); 2Department of Veterinary Sciences, School of Agricultural and Veterinary Sciences, UTAD, 5000-801 Vila Real, Portugal; 3Associate Laboratory for Animal and Veterinary Science (AL4AnimalS), 1300-477 Lisboa, Portugal; 4Centre for Applied Photonics, INESC TEC, Faculty of Sciences of University of Porto, Rua do Campo Alegre, 4169-007 Porto, Portugal; jmmma@gmail.com (F.M.-S.); jmmma@utad.pt (J.M.M.M.D.A.); 5Department of Physics, School of Sciences and Technology, UTAD, 5000-801 Vila Real, Portugal

**Keywords:** antioxidant activity, beef, essential oil, *Myrtus communis* L., minimum inhibitory concentration, spoilage microbiota, active packaging

## Abstract

The aim of this study was to assess the antimicrobial effects of myrtle (*Myrtus communis* L.) essential oil (EO) on pathogenic (*E. coli* O157:H7 NCTC 12900; Listeria monocytogenes ATCC BAA-679) and spoilage microbiota in beef and determine its minimum inhibitory concentration (MIC) and antioxidant activity. The behavior of LAB, Enterobacteriaceae, *Pseudomonas* spp., and fungi, as well as total mesophilic (TM) and total psychotropic (TP) counts, in beef samples, was analyzed during storage at 2 and 8 °C in two different packaging systems (aerobiosis and vacuum). Leaves of myrtle were dried, its EO was extracted by hydrodistillation using a Clevenger-type apparatus, and the chemical composition was determined using chromatographical techniques. The major compounds obtained were myrtenyl acetate (15.5%), β-linalool (12.3%), 1,8-cineole (eucalyptol; 9.9%), geranyl acetate (7.4%), limonene (6.2%), α-pinene (4.4%), linalyl o-aminobenzoate (5.8%), α-terpineol (2.7%), and myrtenol (1.2%). Myrtle EO presented a MIC of 25 µL/mL for *E. coli* O157:H7 NCTC 12900, *E. coli*, *Listeria monocytogenes* ATCC BAA-679, Enterobacteriaceae, and *E. coli* O157:H7 ATCC 35150 and 50µL/mL for *Pseudomonas* spp. The samples packed in aerobiosis had higher counts of deteriorative microorganisms than samples packed under vacuum, and samples with myrtle EO presented the lowest microbial contents, indicating good antimicrobial activity in beef samples. Myrtle EO is a viable natural alternative to eliminate or reduce the pathogenic and deteriorative microorganisms of meat, preventing their growth and enhancing meat safety.

## 1. Introduction

Pathogenic and food spoilage bacteria have been considered the primary causes of foodborne diseases and food quality deterioration in both developed and developing countries [1]. Decontamination treatments or the addition of chemical preservative agents to food products have been widely applied in food industries to assure food safety and extend the shelf life of food products [2]. However, some of these treatments have already been proven to be ineffective against certain microorganisms since the survival of environment-adapted bacteria after treatment processes may lead to high resistance of bacteria such as Escherichia coli O157:H7 and Listeria monocytogenes [3]. Therefore, plant-based antimicrobials, such as essential oils (EOs), have been elected as potential alternatives to synthetic preservatives besides consumers’ preference for more natural foods and an ecofriendly status [4].

Natural EOs are obtained from aromatic and medicinal plants located mainly in temperate climate areas, such as countries surrounding the Mediterranean [5], which can be extracted from the whole plant, including the peel, leaves, flowers, roots, and seeds [5,6]. EOs are organic compounds of a low molecular weight and a complex constitution that can contain up to 60 different compounds, when one-dimensional chromatographic techniques are used, or more, when heart-cut or two-dimensional chromatographic techniques are used. They typically comprise three main groups, such as terpenes, aromatic compounds, and terpenoids [6]. EOs have been firmly established as one of the best alternatives to reduce the microbiota of meat due to their antioxidant and antimicrobial activities [5,7], as well as their medicinal, fungicidal, bactericidal, and antiviral properties [6]. Moreover, several studies have been conducted on the use of EOs as antimicrobials, showing that they increase the safety and shelf-life of food products in addition to their use as flavoring agents in foods. The main mechanism of action of EOs against bacteria may be due to their ability to penetrate through the bacterial membrane into the cell, exhibiting an inhibitory activity on the functional cell [8]. Gram-positive bacteria are more susceptible to this effect than Gram-negative ones, as the latter possesses an outer lipo-polysaccharide layer restricting the flow rate of lipophilic EOs into the intracellular environment [9]. *Acinetobacter*, *Alteromonas*, *Aeromonas*, *Moraxella*, *Flavobacterium*, *Leuconostoc*, and *Pseudomonas* genera are bacteria prevalent in meat and meat products that are responsible for deterioration, which also include Lactic acid bacteria (LAB), *Brochothrix*, and different genera of the Enterobacteriaceae family [10]. Moreover, some studies mention that antimicrobial activity might be correlated with phenolic content [9], while others suggest that oxygenated compounds might be major contributors to this effect [11]. 

Evidence from published assays demonstrates that EO’s antioxidant activity might be due to the presence of phenolic compounds in their constitution, including carvacrol, eugenol, geraniol, chavicol, menthol, linalool, citronellol, and thymol in larger concentrations [5]. Nevertheless, it is important to consider that, as pointed out by different assays, the bactericidal and antioxidant effects cannot be solely attributed to the predominant compounds. Equally significant is the interplay between these major compounds and those present in lower concentrations [9,12,13]. Phenolic compounds act as magnets of free radicals formed in lipid oxidation, donating an electron to the free radical. The phenolic ring reduces the energy of the free radical so that the resulting radicals of this connection (phenolic compounds with free radicals) react with further radicals, spawning non-radical species (non-reactive). These compounds also act as inhibitors of enzymatic systems responsible for initiating meat oxidation reactions [5]. However, EOs can have some limitations when inoculated in the matrix (meat) because there are interactions between the EO and meat structure (lipids, carbohydrates, salts, and proteins) that reduce the antimicrobial and antioxidant effects [5]. 

*Myrtus communis* L., or myrtle, is a small evergreen shrub with dense foliage belonging to the Myrtaceae family, and it is one of the most important medicinal and aromatic species, which is used as a spice for meat dishes, mainly due to its pleasant aroma [14,15]. Their essential oil can be extracted from various plant organs, such as the leaves, berries (round and dark blue), flowers, and roots, and it is typical of Mediterranean countries [15]. The leaves from myrtle are the most obvious part of the plant for EO extraction as they are the most often used as a flavoring and food seasoning [15]. Myrtle EO can be classified into the cineoliferum type, rich in terpenes (α-pinene, limonene, and terpenoid oxides (1,8-cineole)), and the myrtenilacetatiferum type, rich in terpenic esters (terpenil acetate, linalyl-acetate, and bornyl-acetate) and terpenoid oxides (1,8-cineole) [16]. It is characterized by a pleasant aroma and taste, in which α-pinene, 1,8-cineole, limonene, myrtenyl acetate, and myrtenol are the main contributors [17,18]. Myrtle properties represent an added value at an industrial level, as they have been shown to have antimicrobial and antioxidant activities [17]. It has been demonstrated that they also have anti-diabetic, anti-inflammatory, and anti-genotoxic properties, probably due to the contribution of phenolic acids (i.e., caffeic acid) and flavonoids (i.e., myricitrin), among others [14]. According to Aleksic and Knezevic [19], 1,8-cineole, β-linalol, eugenol, and α-terpineol present in myrtle EO have a bactericidal effect. 

In general, terpenes, terpenoids, and phenylpropanoids contribute to the diversity of the antimicrobial and antioxidant properties of essential oils obtained from plants. Terpenes are hydrocarbons synthesized in the cytoplasm of plant cells from isoprene units, while terpenoids are terpenes that undergo further modifications through enzymatic processes by adding oxygen molecules and removing methyl groups. Phenylpropanoids are synthesized from the amino acid precursor phenylalanine in plants and this has a six-carbon aromatic phenol group and a three-carbon propene tail derived from cinnamic acid [19]. 

Research on the impacts of EOs on food products and their biological activities including antimicrobial activity is still lacking. Hence, the present study, which focuses on the current knowledge about the role of EOs in food preservation and their antimicrobial behavior and antioxidant activities, is innovative. Our objective is to study the antimicrobial effect of EOs extracted from *Myrtus communis* L. leaves at the level of microbiota isolated from decaying fresh “Maronesa” beef, a Portuguese autochthonous breed of mountain cattle, determining their minimum inhibitory concentration (MIC) and chemical composition using chromatographical techniques. 

## 2. Materials and Methods

### 2.1. Experimental Design

#### 2.1.1. EO Extraction

Fresh aerial parts of *Myrtus communis* L. naturally occurring in northern Portugal were collected in Vila Real. The specimen was properly and botanically identified. The leaves of *Myrtus communis* L. were dried at a temperature of 40 °C and submitted to hydrodistillation for 3 h using a Clevenger-type apparatus (1:10 with water) [20]. The EO was separated from the water and adequately stored in the dark at −20 °C.

#### 2.1.2. EO Volatile Profile 

The volatile composition of *Myrtus communis* L. EO was obtained using gas chromatography-mass spectrometry (GC-MS) and gas chromatography-thermal conductivity detection analysis. Chromatographical analysis was carried out using a Thermo Scientific™ TRACE™ 1300 Gas Chromatograph coupled to an ISQ™ Series Single Quadrupole MS (Thermo Fisher Scientific Inc.—Bremen, Germany). Analyte separation was performed with a Thermo Scientific TG-5MS column (60 m × 0.25 mm × 0.25 µm). The oven temperature program was initiated with a temperature of 60 °C, held for 2 min, increasing to 280 °C at a rate of 10.00 °C/min and then held for 5 min. Samples and standards were prepared prior to analysis using n-Hexane (Merck—Darmstadt, Germany) in 1.0% (*v*/*v*) and 0.2% (*v*/*v*) concentrations, respectively, and the volume injected was 1.0µL using an autosampler. The injector was set to split mode (1:5), operating at 250 °C and 165 kPa. The mass spectrometer transfer line and ion source temperatures were set to 280 °C and 250 °C, respectively, with the last operating under electron impact mode (70 eV, mass scan range of 30–400 amu). Analysis of the same samples was simultaneously executed on a Shimadzu™ GC-2010 Plus (Shimadzu Corporation—Kyoto, Japan). Separation of analytes was performed with a Zebron ZB-5 column (30 m × 0.25 mm × 0.25 µm) using a similar oven temperature program and injection/injector parameters, except for carrier gas flow, which was set to 82.5 kPa. The detector temperature and current were programmed to 300 °C and 75 mA, respectively, with a make-up flow of 5.0 mL/min. All analytical separations were performed using Helium 99.9995% as carrier gas. Identification of analytes was performed via a comparison of Kovats and linear retention indices, using NIST/EPA/NIH Mass Spectral Library (2011) and other libraries, namely, Pherobase, and by a comparison of authentic standards.

#### 2.1.3. Antioxidant Activity of EO

The determination of the antioxidant activity of EO was performed by implementing a Trolox equivalent antioxidant capacity (TEAC)-adapted method [21], which assesses the total radical-scavenging capacity based on the ability of a compound to scavenge the stable ABTS radical (ABTS^•+^) [22]. To a previously prepared 6.93 mM ABTS^•+^ solution, 88 µL of a potassium persulfate (140 mM) solution was added and allowed to react, in the absence of light, for 12–16 h. Afterward, a calibration curve was obtained by adding to 2 mL aliquots of ABTS^•+^ solution different volumes of Trolox (0.11 mM), ranging from 25 to 200 µL, with total volume correction by adding water. All spectrophotometric measurements were performed by measuring absorption at λ_max abs_ = 734 nm. In order to determine the inhibition value, 25 µL of *Myrtus communis* L. EO was added under constant stirring to 2 mL aliquots of ABTS^•+^ solution and allowed to stabilize for 15 min prior to the spectrophotometric measurements. All measurements were performed in triplicate, and the final value was obtained by arithmetic average.

#### 2.1.4. EO Antibacterial Activity

The minimum inhibitory concentration (MIC) aims to verify/quantify if the EO is operating on the matrix with the desired antimicrobial effect [9]. The method for MIC determination was performed on 96-well sterile microtiter plates according to the method of Sarker et al. [23]. *E. coli* O157:H7 ATCC 35150, *E. coli* O157:H7 NCTC 12900, and *L. monocytogenes* ATCC BAA-679 isolates, as well as isolates of Pseudomonas spp., *E. coli*, and Enterobacteriaceae, obtained from “Maronesa” fresh meat samples were incubated at growth-optimal temperatures for 24 h in brain–heart infusion (BHI) for further subculture medium. The inoculum was prepared with saline solution in order to obtain a concentration of 5 × 10^6^ cfu/mL using the McFarland level [24] and spread on a Petri dish with selective media for each microorganism. Sterile BHI was used as the matrix in a 96-well microtiter plate (100 µL/well), and 100 µL of EO was added to columns 4 through 12 (triplicates). A dilution series of the EO was obtained through geometric and successive dilutions (50% decrease in concentration per well/line advance). The positive control was obtained with the BHI and bacterial suspension (5:1), while the negative control was obtained with BHI alone. Finally, the bacterial suspension and resazurin were consecutively added (20 µL to each well) to the remaining columns. Plates were incubated at growth-optimal temperature for 24–48 h, wherein each microorganism was uniquely and individually tested. The highest dilution with no visible growth was considered the MIC value.

#### 2.1.5. Microbial Analysis 

Fresh beef from semitendinosus and semimembranosus muscles of DOP-Maronesa breed (males; n = 4) were obtained from a local market and transported to the laboratory. After that, 20 g of each sample was individually packed in duplicate in two different conditions: aerobiosis and vacuum, with and without EO (control). Samples were stored at 2 and 8 °C and analyzed at 0, 1, 24, 40, 72, 84, 168, 240, and 336 h for aerobiosis and 0, 1, 24, 168, 336, 504, 671, and 840 h for vacuum package. The microorganisms analyzed were Lactic acid bacteria (LAB), Enterobacteriaceae, *Pseudomonas* spp., and fungi, as well as total mesophilic (TM) and psychotropic (TP) counts. The collection and weighing of the samples for evaluation in each storage temperature was carried out by aseptically removing 10 g of each sample, which was diluted in 90 mL of tryptone salt solution (0.3% tryptone and NaCl at 0.85%, sterilized at 121 °C for 15 min) and homogenized in a “stomacher” for 30 s. Successive decimal dilutions were performed in test tubes containing 9 mL sterile tryptone salt. Afterward, they were sown by incorporation or at the surface depending on the microorganism and the culture medium. Colony counting results were expressed as log cfu/g. For TM [25] and TP [26], spreading was achieved via the incorporation of 1 mL of the original suspension and the respective dilutions in plate count agar (PCA), and spread plates were incubated at 30 °C for 72 h for the TM and 7 °C for 10 days for TP. For Enterobacteriaceae [27], spreading was performed via the addition of 1 mL of the original suspension and the respective dilutions to Violet Red Bile Glucose Agar (VRBG) selective medium (Scharlau 01-295-500) with a double layer. The plates were placed at 37 °C, and after 24 h, counting of typical colony was performed. According to [27], 5 colonies peaked and transferred to nutrient agar and placed at 30 C for 24 h for biochemical confirmation. For *Pseudomonas* spp. enumeration [27], spreading was achieved via the incorporation of 1 mL of the original suspension and the respective dilutions in cephalothin–sodium fusidate–cetrimide (CFC) agar (OXOID CM0559) and CFC selective supplement (OXOID SR0103). The spread plates were incubated at 25 °C for 72 h. After colony counting, 5 were transplanted to nutrient agar and incubated at 30 °C for 24 h, then subjected to an oxidase test according to the method described by the International Standard [28]. LAB [29] spreading was conducted via the addition of 1 mL of the original suspension and the respective dilution to double-layer selective medium Man–Rogosa–Sharpe Agar (MRS) (Oxoid CM0361). The inoculated plates were incubated at 30 °C for 72 h. For fungi [30], spreading was achieved on the surface of 0.1 mL of the original suspension and the respective dilutions through selective culture Glucose Chloramphenicol Agar (GCA) (VWR 84604.0500).

### 2.2. Data Analysis

The effect of EO and time of packaging were evaluated through ANOVA one-way with the STATISTIC 2014 SW program (StatSoft Inc., Tulsa, OK, USA), in which the effect was considered not significant when *p* ≥ 0.05; significant when *p* < 0.05; very significant when *p* < 0.01; and highly significant when *p* < 0.001, and determined by Tukey’s HSD test (“Honestly Significantly Different”).

## 3. Results

The yield of EO extracts of *Myrtus communis* L. was 0.57% (*v*/*w*) based on the dry weight of the plant. The chemical profile of myrtle EO revealed 32 volatile compounds (representing 83.2% of the total content), classified into the three following categories: terpenes (13.0%), terpenoids (68.0%), and phenylpropanoids (2.2%), which were also present but in low quantities. The main compounds of myrtle EO were ethyl myrtenyl (15.5%), β-linalool (12.3%), 1,8-cineole (9.9%), ethyl geranyl (7.4%), limonene (6.2%), α-pinene (4.4%), linalyl o-aminobenzoate (5.6%), α-terpineol (2.7%), α-terpinyl acetate (2.2%), methyl eugenol (1.8%) and myrtenol (1.2%), propanoic acid,2-methyl-,2- (1.8%), 7-isopropyl-7-methyl-nona-3,5-diene-2,8-dione (1.7%), humulene-1,2-epoxide (1.2%), and trans-pinocarvyl acetate (1.2%). The remaining volatile compounds showed quantities below 1%, namely, 2-hexenal, nonane, β-myrcene, terpinen-4-ol, nonane, 2,6-octadienoic acid,3,7-dimethyl-, β-caryophyllene, humulene, and caryophyllene oxide (Appendix A, Appendix A). The antioxidant activity of myrtle EO was 5.37 μmol Trolox/g sample, and this result may be due to the percentage of volatile compounds with antioxidant activity, such as linalool and limonene.

The minimum inhibitory concentration (MIC) values obtained for *Myrtus communis* L. EO for pathogenic bacteria and bacterial isolates in “Maronesa” fresh beef were 25 µL/mL for *E. coli* O157:H7 NCTC 12900, Listeria monocytogenes ATCC BAA-679, *E. coli*, LAB, and Enterobacteriaceae and 50 µL/mL for *E. coli* O157:H7 ATCC 35150 and *Pseudomonas* spp. 

Table 1 and Table 2 present TM, TP, Enterobacteriaceae, LAB, *Pseudomonas* spp., and fungi counts (cfu/g, mean and standard deviation) in beef in an aerobiotic atmosphere over the storage period (336 h) at two different temperatures (2 °C and 8 °C).

Significant differences were observed between control and samples with the presence of myrtle EO for TM at 72 h and for TP at 84 h, both until the end of storage. To highlight this, in the case of Enterobacteriaceae in samples with EO, the counts were 3.7 cfu/g compared to 6.2 cfu/g in the control samples (*p* < 0.05). *Pseudomonas* spp. was significantly affected by the presence of EO from 48 h until the end of storage.No significant differences were observed between the control and the samples with the presence of myrtle EO after 1 and 24 h of inoculation. However, after one hour of myrtle EO inoculation in beef, the counts of all microorganisms decreased. In the case of TP and *Pseudomonas* spp., significant differences were observed between 48 and 336 h of storage, with EO presenting consistently lower counts. Other significant differences were observed for fungi at 48 and 72 h; LAB at 48, 240, and 336 h; and TM from 84 h to the end of storage, with myrtle EO being generally beneficial as it reduced microbial development.

The following figures represent TM (Figure 1), TP (Figure 2), Enterobacteriaceae (Figure 3), LAB (Figure 4), *Pseudomonas* spp. (Figure 5), and fungi (Figure 6) counts (cfu/g, mean and standard deviation) in beef in aerobiosis over the storage period according to two different temperatures.

Storage at high temperatures (8 °C) in aerobiosis revealed high counts of Enterobacteriaceae in both the control and in the presence of myrtle EO. Particularly, at 2 °C, the antibacterial effect of myrtle EO was very evident, keeping Enterobacteriaceae counts below 4 cfu/g, even at 360 h of storage. In the presence of EO, lower counts were obtained for TM and TP in aerobiosis packaging. After 72 h of storage control, the samples at a 2 °C storage temperature attained higher counts than the EO samples at an 8 °C storage temperature, demonstrating the bacteriostatic effect of myrtle EO at higher temperatures (8 °C). Moreover, the presence of myrtle EO also revealed action against fungi, *Pseudomonas* spp., and LAB, which were more evident at a 2 °C storage temperature. With the exception of Enterobacteriaceae, at both temperatures, the samples with EO presented a reduction in microbial load in the first hour of storage. 

Table 3 and Table 4 present the TM, TP, Enterobacteriaceae, LAB, *Pseudomonas* spp., and fungi counts (cfu/g, mean and standard deviation) in beef in a vacuum atmosphere over the storage period (840 h) at two different temperatures (2 °C and 8 °C).

Significant differences were observed between the myrtle EO samples and control samples during the storage time. With the exception of Enterobacteriaceae, at both temperatures, the samples with EO presented a reduction in the microbial load in the first hour of storage. At 24 h of storage, for TM, a significant difference was observed between the EO and control samples, which remained until the end of the storage period (840 h), with EO obtaining consistently lower counts. No significant differences were observed between the control samples and the samples with the presence of myrtle EO for Enterobacteriaceae and fungi. Moreover, no significant differences were observed for EO samples for Enterobacteriaceae and Pseudomonas spp. during all storage times, or for both the control and EO samples in the case of fungi. It should be noted that similar counts were obtained throughout the entire storage period for Enterobacteriaceae, with EO counts of 1.3 cfu/g (1 h after inoculation) and 1.6 cfu/g (840 h after inoculation), and for *Pseudomonas* spp., with EO counts of 4.3 cfu/g (1 h after inoculation) and 4.7 cfu/g (840 h after inoculation). In the control samples at 840 h of storage, higher counts of 5.1 cfu/g for Enterobacteriaceae and 6.6 cfu/g for *Pseudomonas* spp were attained. It should be noted that samples with myrtle EO have a lower microbial load in all vacuum-packed samples stored at 2 °C, reaching an average of 7.20 log cfu/g for TM at the end of the storage time, which is an acceptable limit.

Significant differences were observed for *Pseudomonas* spp. after 24 h and until the end of storage period, and in the case of fungi, significant differences occurred from 168 h to 840 h of storage. At both temperatures, with the exception of Enterobacteriaceae, a reduction in the microbial load was observed in the first hour of storage in samples with EO. No significant differences were observed between the control samples and the samples with the presence of myrtle EO for TM, Enterobacteriaceae, and LAB. As already mentioned, the sample with myrtle EO showed significant differences in relation to the control sample, being generally beneficial, as it reduces microbial development. 

The following figures represent TM (Figure 7), TP (Figure 8), Enterobacteriaceae (Figure 9), LAB (Figure 10), *Pseudomonas* spp. (Figure 11), and fungi (Figure 12) counts (cfu/g, mean and standard deviation) in beef in vacuum over the storage period according to two different temperatures.

In vacuum packaging, the presence of EO revealed the inhibition of Enterobacteriaceae development, with counts below 2 and 3 cfu/g at 840 h of storage, at 2 and 8 °C, respectively. Good performance of myrtle EO was obtained for Pseudomonas spp. at 2 °C, with a count of 4.7 ± 0.3 log cfu/g at 840 h. Moreover, at 2 °C storage, a reduction of one logarithmic unit was observed for Pseudomonas spp. counts from 1 h to 24 h. For fungi, in samples stored under vacuum at 2 °C, there was an increase in fungal counts of approximately one logarithmic unit from the initial time to 840 h. At 8 °C, the control samples showed an increase of one logarithmic unit compared to samples with myrtle EO. It should also be noted that, similar to the behavior of other microorganisms, there was the inhibition of multiplication in the samples with EO.

## 4. Discussion

The EO of *Myrtus communis* L. was studied based on the chemical composition and antimicrobial and antioxidant activities. 

The yield of essential oil extract of *Myrtus communis* L. was 0.57% (*v*/*w*) based on the dry weight of the plant, which is similar to that attained in other studies [19,31] (0.55% and 0.69% (*v*/*w*), respectively) but considerably lower than that obtained by Cherrat et al. [32] (0.89% (*v*/*w*), which might be explained by geographical variability. The chemical profile of the myrtle EO revealed 32 volatile compounds (representing 83.2% of the total content) and was classified according to the chemical composition described by several authors [15,29]: terpenes represent 13.0%, of which 11.4% is monoterpenes and 1.4% is sesquiterpenes; terpenoids are about 68.0%, 17.5% is alcohols, 36.1% is esters, 13.7% is ethers, and 1.2% is aldehydes; lastly, phenylpropanoids represent 2.2% of the EO. Reviewing the literature, it is noticeable that there are two main standpoints on what the main constituents of myrtle EO are: those whose studies conclude that 1,8-cineole and α-pinene are the main constituents [16,18,31,33] and those whose studies point to myrtenyl [17,18,34]. Undoubtedly, it is possible to say that both are correct; whereas 1,8-cineole and α-pinene are predominant in Greece, Italy, France, and Algeria, myrtenyl acetate is prevalent in Portugal, Morocco, Spain, Tunisia, and Albania. In our case, our samples were collected from the Trás-os-Montes region in (northern) Portugal, and their most representative compounds are as follows: myrtenyl acetate (15.5%), β-linalool (12.3%), 1,8-cineole (9.9%), geranyl acetate (7.4%), limonene (6.2%), and α-pinene (4.4%). To a lesser extent, there was also α-terpineol (2.7%), methyl eugenol (1.8%), and myrtenol (1.2%). Comparing the chemical profile of the present study with those attained by other studies, it is noticeable that these values are generally lower than those of [15,32], except for myrtenol (1.2%) and methyl eugenol (1.8%) regarding the latter study. Moreover, geranyl acetate, which is a considerably representative compound in our case, was not mentioned in either case. These compounds have high importance because they have antimicrobial and antioxidant properties, as mentioned previously. Based on our results, myrtle EO contains a high percentage of terpenoids and terpenes, as well as phenylpropanoids at a lower percentage. Among these, alcohols, aldehydes, and phenols were identified. From the total amount of compounds, it was possible to identify that terpenoids represented 68.0%, and of these, a bit more than half were esters (36.1%), a little over a quarter were alcohols (17.1%), and the remaining was mainly ethers (13.70%). A relatively low amount of phenylpropanoids was identified (2.2%). In our work, methyleugenol was identified at a percentage of 1.8. This compound has antioxidant and antimicrobial properties [19]. The MIC value for *Myrtus communis* L. EO relative to pathogenic bacteria and bacteria isolates obtained for *E. coli* O157:H7 NCTC 12900, *E. coli*, Listeria monocytogenes ATCC BAA-679, and Enterobacteriaceae was 25 µL/mL, and for *E. coli* O157:H7 ATCC 35150 and *Pseudomonas* spp., it was 50 µL/mL. According to the work of Hsouna et al. [17] and Cherrat et al. [32], MIC values for *E. coli* O157:H7 and *E. coli* were 25 µL/mL and 56 µL/mL, respectively, despite being from a different serotype, while in the case of Cherrat et al. [32], the MIC for Listeria monocytogenes CECT 4031 was 11.5 µL/mL. However, these results are always dependent on the individual chemical profile of the studied EO, as their composition is variable, and this variability is, per se, dependent on intrinsic factors, such as weather, plant age, the cycle phase of the plant, the organ used for extraction, and the composition of the soil, and operational factors (i.e., the extraction method). According to the obtained results, it is reasonable to infer that Gram-positive bacteria are more susceptible to the antimicrobial activity of the EO. This is a logical assumption since a more direct interaction of the hydrophobic components of EO with the cell membrane is possible, while Gram-negative bacteria are more resilient as they possess a hydrophilic cell wall that acts as an effective hydrophobic barrier. On the other hand, one should not exclude other contributing factors. For instance, terpenes are known to have the ability to disrupt the lipid structure of the cell wall of bacteria, effectively penetrate it, and denature proteins, subsequently leading to the destruction of the cell membrane, thus causing cell lysis and cell necrosis [35]. However, according to [36], over time, EO effects will be the same on both types of bacteria. This can be supported by the results obtained for Enterobacteriaceae in the present study.

The antioxidant activity may be due to the presence of compounds such as B-linalool, geranyl acetate, limonene, α-terpineol, 1,8-cineole (eucalyptol), α-pinene, and methyl eugenol [19]. In our assays, the antioxidant activity of *Myrtus communis* L. EO exhibited a value of 5.37µmol/g, which is lower than those obtained by Moura et al. [37] for *Laurus nobilis* L. (18.86 µmol/g), *Citrus Sinensis* L. (6.74 µmol/g), and [38] *T. algeriensis* (thymus) EO (11.69 µmol/g). From these EOs, and according to our in vitro studies, it can be observed that *Myrtus communis* L. EO is the one with the lowest antioxidant activity, but when it was inoculated on the matrix samples, it showed good performance. 

The behavior of LAB, Enterobacteriaceae, *Pseudomonas* spp., Fungi, TM, and TP in beef samples was analyzed during storage at 2 and 8 °C under two different atmospheres (aerobiosis and vacuum). Samples packed in aerobiosis had higher counts of deteriorative microorganisms than samples packed under vacuum, which was expected. This can be justified by the fact that vacuum packaging is effective at microbial reduction, which is due to the lack of oxygen available [38]. It was also verified that the EO of myrtle had a positive effect on the reduction in the microbial load, mainly at 2 °C. There was some impact of EOs on the development of LAB, compared to that observed in the control samples, which can be explained by the superior antimicrobial activity of EOs against Gram-positive bacteria [39]. In fresh meat packed in aerobiosis, aerobic bacteria are the predominant ones in the matrix, i.e., Pseudomonas spp., consuming the glucose available in the tissues [40]. When these packages are subjected to low refrigeration temperatures, Pseudomonas spp. develop; when subjected to higher cooling temperatures, it leads to the development of Enterobacteriaceae [10]. A different deteriorative pattern was observed with vacuum packaging compared to aerobiosis packaging. It is noteworthy that there was a much higher multiplication of LAB than *Pseudomonas* spp., which can be explained by the fact that an increase in the multiplication of LAB has a beneficial effect on the acidification of meat, that is, on decreasing pH, due to the formation of lactic acid. Pseudomonas spp. decrease their multiplication with a decrease in pH [41] and, essentially, in the absence of oxygen. An effect of myrtle EO was observed on the reduction in Enterobacteriaceae at 2 and 8 °C, maintaining their counts, on average, below 3 log cfu/g in these samples. 

The samples with myrtle EO were the ones with the lowest microbial contents, indicating this EO, when interacting with the matrix, presents strong antimicrobial activity. 

## 5. Conclusions

Myrtle is a shrub that grows spontaneously in Mediterranean areas. In this work, the chemical composition of myrtle EO obtained in the north of Portugal, as well as its antioxidant and antimicrobial activities, was studied. *Myrtus communis* L. EO presented a high percentage of both terpenes and terpenoids, along with phenylpropanoid compounds at a lower percentage. From a chromatographical point of view, it was possible to identify 83.2% of the EO, of which 81.1% may potentially have antimicrobial activity (terpenes and terpenoids), 2.2% is phenylpropanoids, and no phenol-related compounds were identified. For antioxidant activity, myrtle EO was assessed at 5.37 µmol/g, a higher value than that obtained for the in vitro model. Regarding its antimicrobial properties, *Myrtus communis* L. EO showed a MIC of 25 µL/mL for *E. coli* O157:H7 NCTC 12900, *E. coli*, *Listeria monocytogenes* ATCC BAA-679, Enterobacteriaceae, and *E. coli* O157:H7 ATCC 35150, which is in accordance with other authors.

In summary, the essential oil of *Myrtus communis* L. is a natural, practicable alternative to eliminate or reduce pathogens and spoilage microorganisms on meat and prevent their growth, thus extending meat quality for longer periods of time, increasing its safety, and reducing lipid oxidation of meat, which is due to its antioxidant properties.

However, other studies should be performed to analyze and compare EOs obtained from different myrtle plant specimens under similar and controlled conditions to reduce external variables and understand the influence of certain factors on the chemical profile as well as the antimicrobial and antioxidant properties toward certain strands of bacteria. Other aspects such as the bioavailability of myrtle and its metabolites, as well as its impact on humans consuming foods containing this essential oil, need to be studied. Also, the potential replacement of synthetic antimicrobials with natural alternative agents should be considered in the future.

## Figures and Tables

**Figure 1 foods-12-03390-f001:**
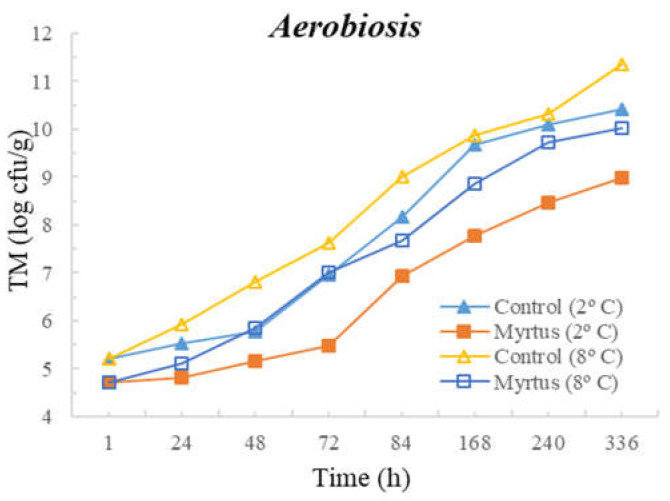
Total mesophilic (TM) counts (cfu/g) over the storage period (2 °C and 8 °C) in air.

**Figure 2 foods-12-03390-f002:**
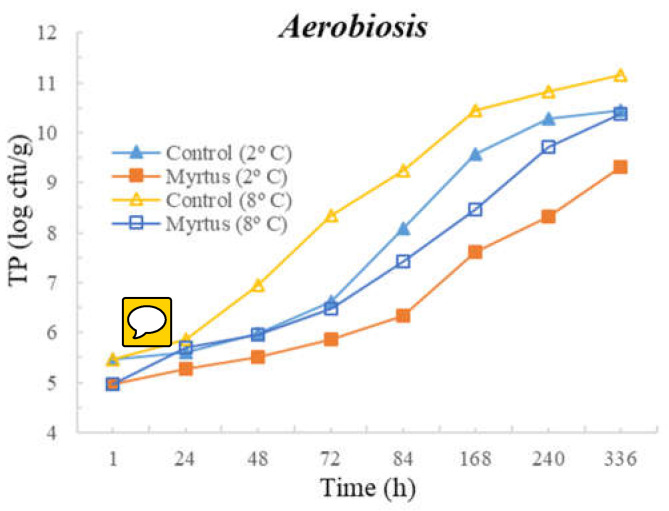
Total psychotropic (TP) counts (cfu/g) in beef over the storage period (2 °C and 8 °C) in air.

**Figure 3 foods-12-03390-f003:**
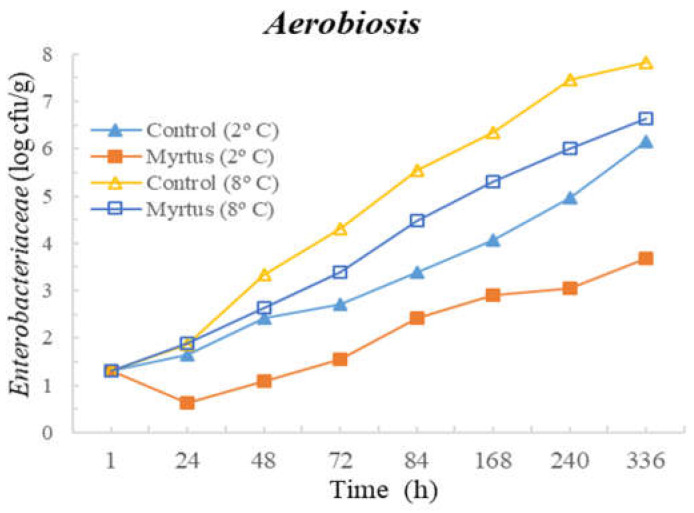
Enterobacteriaceae counts (cfu/g) over the storage period (2 °C and 8 °C) in air.

**Figure 4 foods-12-03390-f004:**
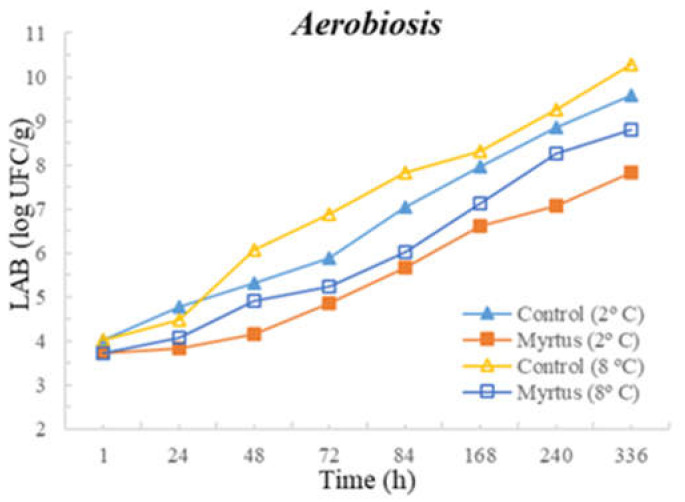
LAB counts (cfu/g) over the storage period (2 °C and 8 °C) in air.

**Figure 5 foods-12-03390-f005:**
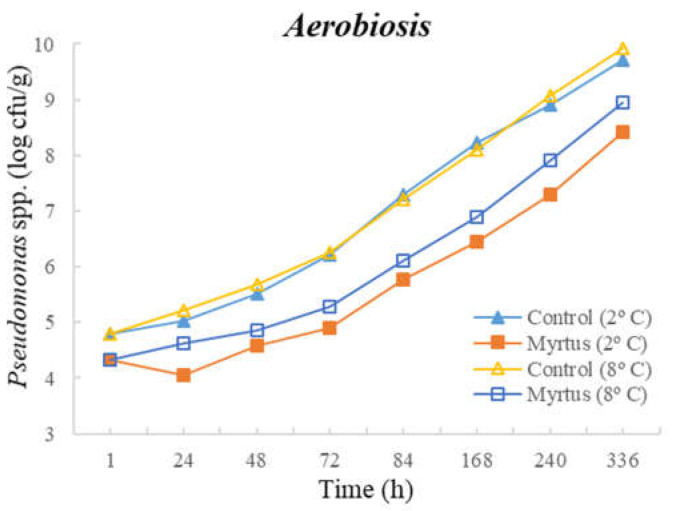
*Pseudomonas* spp. counts (cfu/g) in beef over the storage period (2 °C and 8 °C) in air.

**Figure 6 foods-12-03390-f006:**
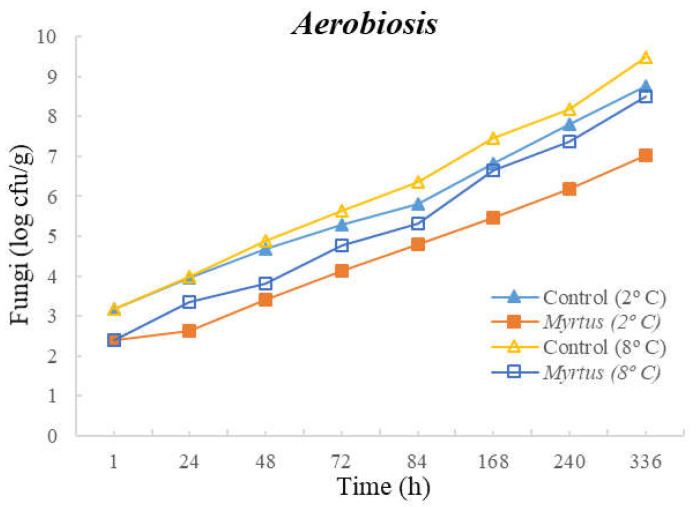
Fungi counts (cfu/g) in beef over the storage period (2 °C and 8 °C) in air.

**Figure 7 foods-12-03390-f007:**
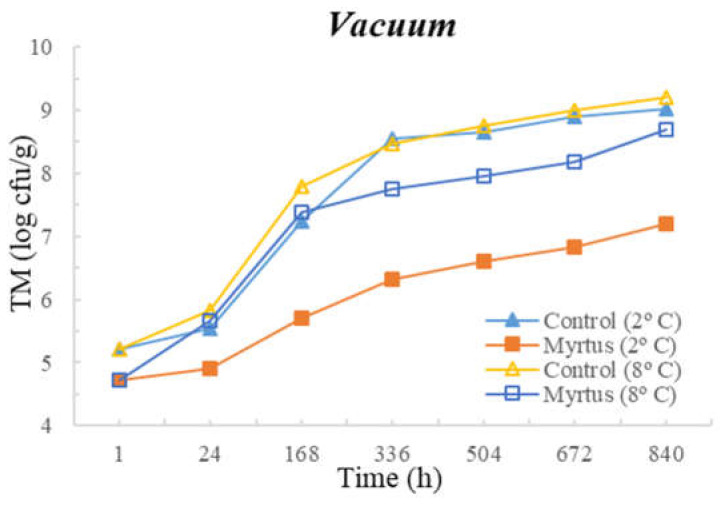
Total mesophilic (TM) counts (cfu/g) in beef over the storage period (2 °C and 8 °C) in vacuum.

**Figure 8 foods-12-03390-f008:**
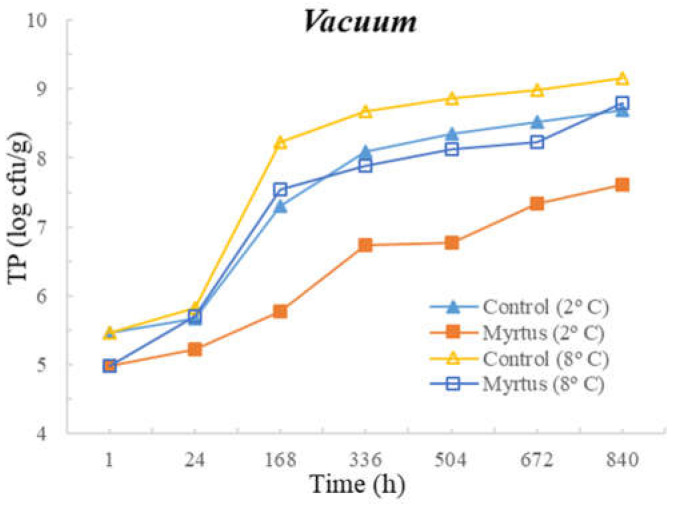
Total psychotropic (TP) counts (cfu/g) in beef over the storage period (2 °C and 8 °C) in vacuum.

**Figure 9 foods-12-03390-f009:**
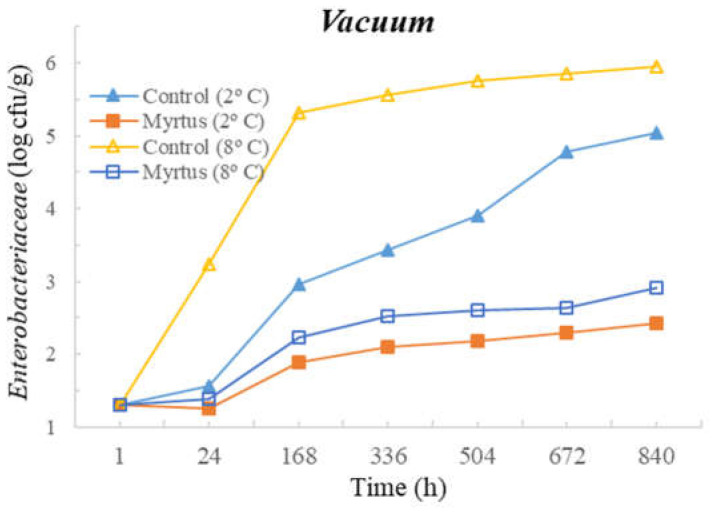
Enterobacteriaceae counts (cfu/g) in beef over the storage period (2 °C and 8 °C) in vacuum.

**Figure 10 foods-12-03390-f010:**
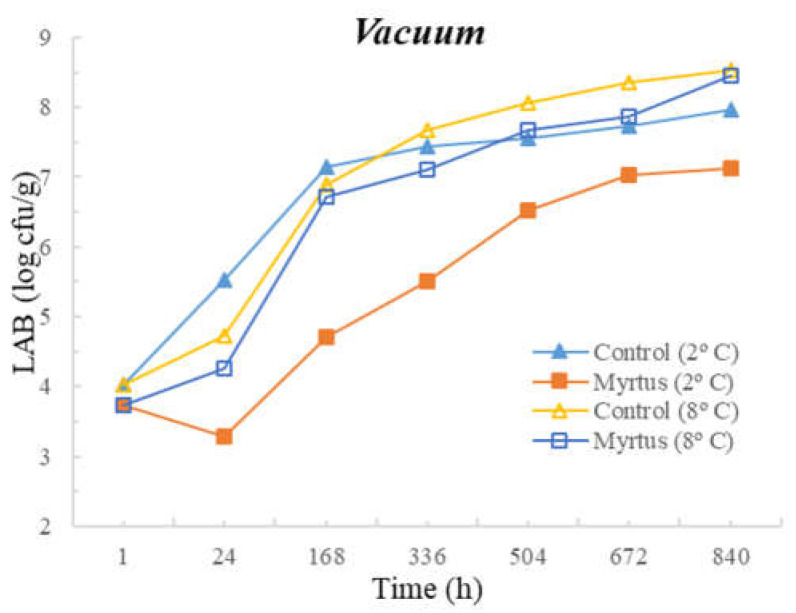
LAB counts (cfu/g) in beef over the storage period (2 °C and 8 °C) in vacuum.

**Figure 11 foods-12-03390-f011:**
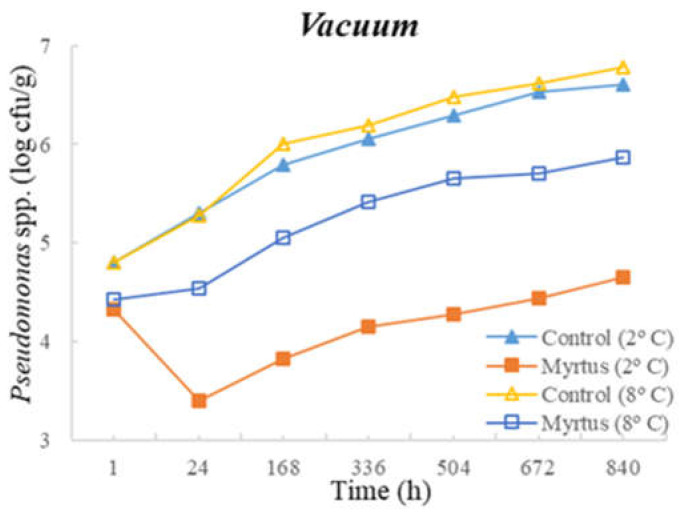
*Pseudomonas* spp. counts (cfu/g) in beef over the storage period (2 °C and 8 °C) in vacuum.

**Figure 12 foods-12-03390-f012:**
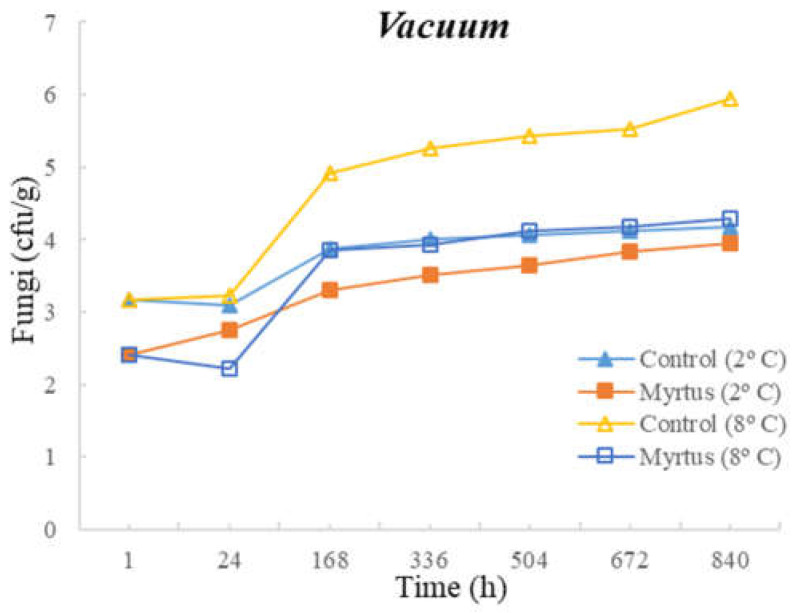
Fungi counts (cfu/g) in beef over the storage period (2 °C and 8 °C) in vacuum.

**Table 1 foods-12-03390-t001:** Mean values ± standard deviation (cfu/g) of deteriorative microbiota counts at 1, 24, 48, 72, 84, 168, 240, and 336 h of storage at temperature of 2 °C.

Aerobiosis	Time (h)	TM			TP			Enterobacteriaceae			LAB			*Pseudomonas* spp.			Fungi		
		Control	EO	Sig	Control	EO	Sig	Control	EO	Sig	Control	EO	Sig	Control	EO	Sig	Control	EO	Sig
**T = 2 °C**	1	5.2 ± 0.8 ^a^	4.7 ± 0.8 ^a^	NS	5.5 ± 0.5 ^a^	5.0 ± 0.3 ^a^	NS	1.3 ± 1.2 ^a^	1.3 ± 1.1 ^ab^	NS	4.0 ± 2.0 ^a^	3.7 ± 1.4 ^a^	NS	4.8 ± 0.9 ^a^	4.3 ± 0.7 ^ab^	NS	3.2 ± 2.9 ^a^	2.4 ± 2.1 ^a^	NS
	24	5.5 ± 0.5 ^ab^	4.8 ± 0.6 ^a^	NS	5.6 ± 0.4 ^a^	5.3 ± 0.1 ^ab^	NS	1.6 ± 1.5 ^a^	0.6 ± 1.1 ^a^	NS	4.8 ± 1.5 ^ab^	3.8 ± 1.8 ^ab^	NS	5.0 ± 0.6 ^a^	4.0 ± 0.4 ^a^	NS	4.0 ± 1.7 ^ab^	2.6 ± 0.8 ^a^	NS
	48	5.5 ± 0.3 ^ab^	5.0 ± 0.2 ^a^	NS	5.7 ± 0.0 ^a^	5.5 ± 0.0 ^ab^	NS	2.3 ± 0.1 ^ab^	1.2 ± 1.2 ^ab^	NS	5.3 ± 0.9 ^abA^	4.1 ± 0.0 ^ab^	*	5.5 ± 0.1 ^a^	4.6 ± 0.2 ^ab^	**	4.7 ± 0.5 ^ab^	3.4 ± 0.3 ^ab^	*
	72	7.0 ± 0.8 ^bc^	5.5 ± 0.5 ^ab^	*	6.6 ± 0.5 ^ab^	5.9 ± 0.1 ^bc^	NS	2.7 ± 0.4 ^abc^	1.5 ± 1.4 ^ab^	NS	5.9 ± 1.4 ^abc^	4.9 ± 1.0 ^abc^	NS	6.2 ± 0.8 ^ab^	4.9 ± 0.4 ^ab^	**	5.3 ± 0.6 ^abc^	4.1 ± 0.2 ^ab^	*
	84	8.2 ± 0.3 ^cd^	6.9 ± 0.5 ^bc^	*	8.1 ± 0.9 ^b^	6.3 ± 0.2 ^c^	*	3.4 ± 0.2 ^abc^	2.4 ± 0.6 ^ab^	NS	7.0 ± 1.0 ^abcd^	5.7 ± 0.6 ^abc^	NS	7.3 ± 0.4 ^bc^	5.8 ± 0.4 ^bc^	*	5.8 ± 0.5 ^abc^	4.8 ± 0.9 ^ab^	NS
	168	9.7 ± 0.6 ^de^	7.8 ± 0.2 ^cd^	**	9.6 ± 0.8 ^c^	7.6 ± 0.5 ^d^	*	4.1 ± 0.2 ^cde^	2.9 ± 0.8 ^ab^	NS	8.0 ± 0.8 ^bcd^	6.6 ± 0.5 ^abc^	NS	8.2 ± 0.7 ^c^	6.4 ± 0.4 ^c^	*	6.8 ± 0.5 ^abc^	5.5 ± 1.1 ^ab^	NS
	240	10.1 ± 0.3 ^e^	8.5 ± 0.4 ^d^	**	10.3 ± 0.2 ^c^	8.3 ± 0.2 ^d^	***	5.0 ± 0.3 ^deA^	3.1 ± 1.0 ^abB^	*	8.9 ± 0.2 ^cd^	7.1 ± 0.5^bc^	NS	8.9 ± 0.5 ^cd^	7.3 ± 0.6 ^cd^	**	7.8 ± 0.5 ^bc^	6.2 ± 1.3 ^ab^	NS
	336	10.4 ± 0.5 ^e^	9.0 ± 0.0 ^d^	**	10.5 ± 0.3 ^c^	9.3 ± 0.1 ^e^	**	6.2 ± 0.8 ^eA^	3.7 ± 0.6 ^bB^	*	9.6 ± 0.3 ^d^	7.8 ± 1.0 ^c^	NS	9.7 ± 0.4 ^d^	8.4 ± 0.4 ^d^	*	8.8 ± 0.2 ^c^	7.0 ± 1.5 ^b^	NS
	Sig	***	***		***	***		***	*		***	***		***	***		**	**	

In each column, means with different letters differ significantly: * *p* < 0.05; ** *p* < 0.01; *** *p* < 0.001; NS: non-significant.

**Table 2 foods-12-03390-t002:** Mean values ± standard deviation (cfu/g) of deteriorative microbiota counts at 1, 24, 48, 72, 84, 168, 240, and 336 h of storage at temperature of 8 °C.

Aerobiosis	Time (h)	TM			TP			Enterobacteriaceae			LAB			*Pseudomonas* spp.			Fungi		
		Control	EO	Sig	Control	EO	Sig	Control	EO	Sig	Control	EO	Sig	Control	EO	Sig	Control	EO	Sig
	1	5.2 ± 0.8 ^a^	4.7 ± 0.8 ^a^	NS	5.5 ± 0.5 ^a^	5.0 ± 0.3 ^a^	NS	1.3 ± 1.2 ^ab^	1.0 ± 0.9 ^a^	NS	4.0 ± 2.0 ^ab^	3.7 ± 1.4 ^a^	NS	4.8 ± 0.9 ^ab^	4.3 ± 0.6 ^a^	NS	3.2 ± 2.9 ^ab^	2.4 ± 2.1 ^a^	NS
**T = 8 °C**	24	5.9 ± 0.5 ^ab^	5.1 ± 0.4 ^a^	NS	5.9 ± 0.7 ^ab^	5.7 ± 0.2 ^abc^	NS	1.9 ± 1.6 ^abc^	1.9 ± 0.3 ^ab^	NS	4.5 ± 2.1 ^cb^	4.1 ± 1.4 ^ab^	NS	5.2 ± 0.7 ^ab^	4.6 ± 0.7 ^a^	NS	4.0 ± 1.7 ^abc^	3.4 ± 0.9 ^ab^	NS
	48	6.8 ± 0.1 ^b^	5.7 ± 0.3 ^a^	NS	6.6 ± 0.2 ^b^	5.9 ± 0.0 ^bc^	*	3.4 ± 0.1 ^cde^	2.7 ± 0.4 ^ab^	NS	6.1 ± 0.2 ^abc^	4.9 ± 0.0 ^ab^	*	5.6 ± 0.2 ^ab^	4.9 ± 0.7 ^a^	**	4.4 ± 0.2 ^abcd^	3.5 ± 0.6 ^ab^	*
	72	7.6 ± 1.2 ^bc^	7.0 ± 0.4 ^b^	NS	8.3 ± 0.3 ^c^	6.5 ± 0.1 ^c^	*	4.3 ± 0.5 ^def^	3.4 ± 0.6 ^bc^	NS	6.9 ± 0.8 ^bcd^	5.2 ± 0.9 ^abc^	NS	6.3 ± 0.2 ^bc^	5.3 ± 0.8 ^ab^	*	5.6 ± 0.6 ^abcd^	4.7 ± 0.8 ^ab^	*
	84	9.0 ± 0.7 ^cd^	7.7 ± 0.3 ^bc^	*	9.3 ± 0.3 ^c^	7.4 ± 0.3 ^d^	*	5.5 ± 0.9 ^efg^	4.5 ± 0.4 ^cd^	NS	7.8 ± 0.3 ^cde^	6.0 ± 0.6 ^abcd^	NS	7.2 ± 0.7 ^cd^	6.1 ± 0.5 ^abc^	**	6.4 ± 0.4 ^bcde^	5.3 ± 0.7 ^abcd^	NS
	168	9.9 ± 0.4 ^de^	8.9 ± 0.2 ^cd^	**	10.5 ± 0.1 ^d^	8.5 ± 0.4 ^e^	*	6.4 ± 0.5 ^fg^	5.3 ± 0.3 ^de^	NS	8.3 ± 0.6 ^cde^	7.1 ± 0.7 ^bcd^	NS	8.1 ± 0.7 ^de^	6.9 ± 0.6 ^bc^	*	7.5 ± 0.2 ^cde^	6.7 ± 0.2 ^bcd^	NS
	240	10.3 ± 0.3 ^de^	9.7 ± 0.2 ^d^	**	10.8 ± 0.2 ^d^	9.7 ± 0.2 ^f^	***	7.5 ± 0.8 ^g^	6.0 ± 0.4 ^de^	*	9.3 ± 0.6 ^de^	8.3 ± 0.3 ^cd^	**	9.1 ± 0.6 ^cf^	7.9 ± 0.7 ^cd^	*	8.2 ± 0.6 ^de^	7.4 ± 0.2 ^cd^	NS
	336	11.4 ± 0.9 ^e^	10.0 ± 0.0 ^d^	**	11.2 ± 0.3 ^d^	10.4 ± 0.3 ^f^	**	7.8 ± 0.7 ^g^	6.6 ± 0.1 ^e^	*	10.3 ± 0.3 ^e^	8.8 ± 0.5 ^d^	*	9.9 ± 0.3 ^f^	8.9 ± 0.4 ^d^	*	9.5 ± 0.6 ^e^	8.5 ± 0.3 ^d^	NS
	Sig	***	***		***	***		***	***		***	***		***	***		***	**	

In each column, means with different letters differ significantly: * *p* < 0.05; ** *p* < 0.01; *** *p* < 0.001; NS: non-significant.

**Table 3 foods-12-03390-t003:** Mean values ± standard deviation (cfu/g) of deteriorative microbiota counts at 1, 24, 168, 336, 504, 672, and 840 h of storage at temperature of 2 °C in vacuum packaging.

Vacuum	Time (h)	TM			TP			Enterobacteriaceae			LAB			*Pseudomonas* spp.			Fungi		
		Control	EO	Sig	Control	EO	Sig	Control	EO	Sig	Control	EO	Sig	Control	EO	Sig	Control	EO	Sig
**T = 2 °C**	1	5.2 ± 0.8 ^a^	4.7 ± 0.8 ^a^	NS	5.5 ± 0.5 ^a^	45.0 ± 0.3 ^a^	NS	1.3 ± 1.2 ^a^	1.3 ± 1.2 ^a^	NS	4.0 ± 2.0 ^a^	3.7 ± 1.4 ^ab^	NS	4.8 ± 0.9 ^a^	4.3 ± 0.7	NS	3.2 ± 2.9 ^a^	2.4 ± 2.1 ^a^	NS
	24	5.5 ± 1.0 ^ab^	4.9 ± 0.8 ^ab^	*	5.7 ± 0.5 ^b^	5.2 ± 1.3 ^ab^	NS	1.6 ± 1.4 ^a^	0.7 ± 1.2 ^a^	NS	5.5 ± 1.0 ^ab^	3.3 ± 1.6 ^a^	NS	5.3 ± 0.4 ^ab^	3.4 ± 0.3	NS	3.1 ± 0.9 ^a^	2.8 ± 0.5 ^a^	NS
	168	7.2 ± 1.0 ^bc^	5.7 ± 0.2 ^abcd^	***	7.3 ± 0.3 ^bc^	5.8 ± 1.1 ^abc^	NS	3.0 ± 0.0 ^ab^	1.9 ± 1.7 ^a^	NS	7.2 ± 0.0 ^ab^	4.7 ± 0.7 ^abcd^	**	5.8 ± 0.4 ^ab^	3.8 ± 0.5	**	3.9 ± 0.3 ^a^	3.3 ± 0.2 ^a^	NS
	336	8.6 ± 0.1 ^c^	6.3 ± 0.2 ^bcde^	***	8.1 ± 0.3 ^bc^	6.7 ± 0.8 ^abc^	*	3.4 ± 0.2 ^ab^	2.1 ± 1.9 ^a^	NS	7.4 ± 0.1 ^ab^	5.5 ± 0.4 ^abcd^	***	6.1 ± 0.2 ^ab^	4.2 ± 0.5	**	4.0 ± 0.3 ^a^	3.5 ± 0.2 ^a^	NS
	504	8,7 ± 0.0 ^c^	6.6 ± 0.1 ^cde^	***	8.4 ± 0.2 ^bc^	6.8 ± 0.8 ^abc^	**	3.9 ± 0.1 ^b^	1.2 ± 2.1 ^a^	NS	7.6 ± 0.1 ^ab^	6.5 ± 0.1 ^bcd^	**	6.3 ± 0.2 ^b^	4.3 ± 0.5	**	4.1 ± 0.4 ^a^	3.7 ± 0.2 ^a^	NS
	672	8.9 ± 0.0 ^c^	6.8 ± 0.2 ^de^	***	8.5 ± 0.0 ^bc^	7.4 ± 0.4 ^bc^	**	4.8 ± 0.2 ^b^	1.3 ± 2.3 ^a^	NS	7.7 ± 0.1 ^ab^	7.0 ± 0.1 ^cd^	**	6.5 ± 0.3 ^b^	4.4 ± 0.5	**	4.1 ± 0.3 ^a^	3.8 ± 0.2 ^a^	NS
	840	9.0 ± 0.0 ^c^	7.2 ± 0.2 ^e^	***	8.7 ± 0.0 ^c^	7.6 ± 0.2 ^c^	**	5.1 ± 0.1 ^b^	1.6 ± 2.8 ^a^	NS	8.0 ± 0.1 ^b^	7.1 ± 0.0 ^d^	***	6.6 ± 0.2 ^b^	4.7 ± 0.3	***	4.2 ± 0.3 ^a^	4.0 ± 0.1 ^a^	NS
	Sig	***	***		***	**		***	NS		**	*		*	NS		NS	NS	

In each column, means with different letters differ significantly: * *p* <0.05; ** *p* < 0.01; *** *p* < 0.00; NS: non-significant.

**Table 4 foods-12-03390-t004:** Mean values ± standard deviation (cfu/g) of deteriorative microbiota counts at 0, 1, 24, 168, 336, 504, 672, and 840 h of storage at temperature of 8 °C.

Vacuum	Time (h)	TM			TP			Enterobacteriaceae			LAB			*Pseudomonas* spp.			Fungi		
		Control	EO	Sig	Control	EO	Sig	Control	EO	Sig	Control	EO	Sig	Control	EO	Sig	Control	EO	Sig
**T = 8 °C**	1	5.2 ± 0.8 ^a^	4.7 ± 0.8 ^a^	NS	5.5 ± 0.5 ^a^	5.0 ± 0.3 ^a^	NS	1.3 ± 1.2 ^a^	1.3 ± 1.1 ^a^	NS	4.0 ± 2.0 ^a^	3.7 ± 1.4 ^a^	NS	4.8 ± 0.9 ^a^	4.4 ± 0.6 ^a^	NS	3.2 ± 2.9 ^a^	2.4 ± 2.1 ^a^	NS
	24	5.8 ± 0.6 ^a^	5.7 ± 1.1 ^ab^	NS	5.8 ± 0.2 ^a^	5.7 ± 1.1 ^a^	NS	3.2 ± 0.7 ^b^	1.7 ± 1.9 ^a^	NS	4.7 ± 1.4 ^ab^	4.3 ± 1.9 ^ab^	NS	5.3 ± 0.4 ^ab^	4.5 ± 0.6 ^ab^	**	3.2 ± 2.9 ^a^	2.2 ± 2.1 ^a^	NS
	168	7.8 ± 0.6 ^b^	7.4 ± 0.8 ^bc^	NS	8.2 ± 0.5 ^b^	7.6 ± 0.3 ^b^	NS	5.3 ± 0.0 ^c^	2.6 ± 2.3 ^a^	NS	6.9 ± 0.5 ^abc^	6.7 ± 0.7 ^abc^	NS	6.0 ± 0.1 ^abc^	5.1 ± 0.3 ^abc^	**	4.9 ± 0.7 ^a^	3.9 ± 0.3 ^a^	*
	336	8.5 ± 0.5 ^b^	7.8 ± 0.2 ^c^	NS	8.7 ± 0.2 ^bcA^	7.9 ± 0.2 ^b^	**	5.6 ± 0.1 ^c^	2.9 ± 2.6 ^a^	NS	7.7 ± 0.2 ^bc^	7.1 ± 0.5 ^bc^	NS	6.2 ± 0.0 ^bc^	5.4 ± 0.2 ^abc^	**	5.3 ± 0.5 ^a^	3.9 ± 0.2 ^a^	*
	504	8.8 ± 0.3^b^	8.0 ± 0.5 ^c^	NS	8.9 ± 0.0 ^bcA^	8.1 ± 0.4 ^b^	*	5.8 ± 0.1 ^c^	1.7 ± 2.9 ^a^	NS	8.1 ± 0.2 ^c^	7.7 ± 0.5 ^c^	NS	6.5 ± 0.0 ^bc^	5.7 ± 0.1 ^abc^	***	5.4 ± 0.4 ^a^	4.1 ± 0.4 ^a^	*
	672	9.0 ± 0.3 ^b^	8.2 ± 0.6 ^c^	NS	9.0 ± 0.1 ^bc^	8.2 ± 0.4 ^b^	NS	5.9 ± 0.0 ^c^	1.7 ± 3.0 ^a^	NS	8.4 ± 0.1 ^c^	7.9 ± 0.7 ^c^	NS	6.6 ± 0.1 ^bc^	5.7 ± 0.2 ^bc^	***	5.5 ± 0.4 ^a^	4.2 ± 0.4 ^a^	*
	840	9.2 ± 0.3 ^b^	8.7 ± 0.7 ^c^	NS	9.2 ± 0.2 ^c^	8.8 ± 0.4 ^b^	NS	6.0 ± 0.0 ^c^	2.6 ± 2.7 ^a^	NS	8.5 ± 0.1 ^c^	8.5 ± 0.6 ^c^	NS	6.8 ± 0.0 ^cA^	5.9 ± 0.1 ^c^	***	5.9 ± 0.0 ^a^	4.3 ± 0.4 ^a^	**
	Sig	***	***		***	***		***	NS		***	***		***	**		NS	NS	

In each column, means with different letters differ significantly: * *p* < 0.05; ** *p* < 0.01; *** *p* < 0.00; NS: non-significant.

## Data Availability

The data used to support the findings of this study can be made available by the corresponding author upon request.

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
