# Peer review of "Antimicrobial Effects and Antioxidant Activity of Myrtus communis L. Essential Oil in Beef Stored under Different Packaging Conditions"

_foods, 2023, doi:10.3390/foods12183390_

Round 1

Reviewer 1 Report

I read your manuscript with the title “Antimicrobial effects and antioxidant activity of Myrtus communis L. essential oil in beef stored under different packaging conditions” with great interest. I consider the results as interesting and well presented. However, I think the sample design (addition of packaging (without vacuum) with essential oil could be to show the direct influence of essential oil on E. coli. Also, it would be nice if they could discuss their research results more intensively. In addition, to add value to your work, it would be nice if you could highlight its innovative character. It would round off your work if you could show which optimization packaging or food producers can draw from your findings or if you could point out further research needs. Therefore, I recommend that your manuscript should undergo minor revisions according to my comments mentioned below prior to acceptance

1-      Keywords: I suggest to add the keyword active packaging

2- The illustrations are sharper, but the chosen colours and symbols still allow for improvements. I would also appreciate it if you would add some of the explanations given to me to the paper, so that the reader is also informed

3-  All in all, I  miss the innovative character or the differentiation from other works. You should point out one of them and highlight what  needs to be investigated in further research or where there is potential for optimisation

please check your manuscript with someone native speaker english

Author Response

The authors thanks the valuable comments and suggestions of the Reviewer that allowed the improvement of manuscript quality. All the revisions are marked in the text.

Reviewer´s comment:

1-      Keywords: I suggest to add the keyword active packaging~

Our reply: Done as required.

Reviewer´s comment:

2- The illustrations are sharper, but the chosen colours and symbols still allow for improvements. I would also appreciate it if you would add some of the explanations given to me to the paper, so that the reader is also informed

Our reply: Amended as required.

Reviewer´s comment:

3-  All in all, I  miss the innovative character or the differentiation from other works. You should point out one of them and highlight what  needs to be investigated in further research or where there is potential for optimisation.

Our reply: The manuscript was improved as required.

Comments on the Quality of English Language

Reviewer´s comment:

please check your manuscript with someone native speaker english.

Our reply: The English language was revised by one native speaker as required.

Reviewer 2 Report

The reviewed paper concerns myrtle essential oil and its antimicrobial and antioxidant activity. The object of this study was also the chemical composition of EO, however this part is the weakest in this manuscript. Before publication this paper needs corrections. The major issues are:

Introduction

Line 62: the statement that essential oils contain “20 to 60” is confusing. There are essential oils with much more components, and the amount of detected compounds depending on technique used for the analysis. Two dimensional methods allow to detect more that 100 components in a single essential oil.

Lines 80-82: the sentence “The EOs show antioxidant activity due to the presence in their constitution of phenolic compounds, being the carvacrol, eugenol, geraniol, chavicol, menthol, linalool, citronellol and thymol the main contributing compounds [5]” is not clear. If the authors write about EO in general, why only the mentioned alcohols are important? Why such compounds as santalol, nerol, zingiberol, sclareol, spathulenol, patchoulol and many others are not included?

And because of this fragment about phenolic compounds and their importance, why the authors choose EO from M. communis in which the most characteristic components we esters not alcohols?

Line 115: the authors mentioned that the aim of their studies was also “chemical composition by chromatographical techniques.” However I cannot see the data concerning this. Table with identified compounds together with retention indices as well as GC chromatogram should be provided even in the supplementary materials!!!

Material and methods

What king od standards were used for GC-MS?

Based on the description of GC-MS conditions I see that the analysis was performed in the standard GC columns not chiral. The question is how did the authors identify D-limonene, since under normal conditions there is no differences between D- and L-limonene (the same mass spectra and retention indices)? The same question concern (-)-trans-pinocarvyl acetate. Are you really sure that it was minus isomer, not plus???!!!! Did you check this?

How was the percentage of compounds in the EO calculated?

Results

The authors should provide the GC-MS data, even in the supplementary materials. I am doubtful about proper identification of EO components.

Did the authors use the positive control in their studies?

Discussion

I am not sure whether the lower yield of essential oil is because of climatic conditions. It could be because of the drying temperature. The authors mentioned in the Material and methods part that plant material was dried in the temperature 40-45 oC. It is a bit too high for aromatic plants. It should be not more than 35 degrees. I understand that sometimes, because of the climatic conditions it is difficult to keep such a temperature, but it should be mentioned.

Author Response

The authors thanks the valuable comments and suggestions of the Reviewer that allowed the improvement of manuscript quality. All the revisions are marked in the text.

Introduction

Reviewer´s comment:

Line 62: the statement that essential oils contain “20 to 60” is confusing. There are essential oils with much more components, and the amount of detected compounds depending on technique used for the analysis. Two dimensional methods allow to detect more that 100 components in a single essential oil.

Our reply: A quick literature search reinforces the authors’ statement regarding the number of typically found components, which stand around 40 to 45, for Myrtus communis L.. Nevertheless, and in agreement to the comment raised by the reviewer, the statement was rephrased in order to become more precise and clearer, mentioning the potential of heart-cut-based techniques.

Reviewer´s comment:

Lines 80-82: the sentence “The EOs show antioxidant activity due to the presence in their constitution of phenolic compounds, being the carvacrol, eugenol, geraniol, chavicol, menthol, linalool, citronellol and thymol the main contributing compounds [5]” is not clear. If the authors write about EO in general, why only the mentioned alcohols are important? Why such compounds as santalol, nerol, zingiberol, sclareol, spathulenol, patchoulol and many others are not included?

And because of this fragment about phenolic compounds and their importance, why the authors choose EO from M. communis in which the most characteristic components we esters not alcohols?

Our reply: Our opinion is totally aligned with the reviewer’s comment; the sentence has been rephrased in order to better convey the correct message:

“Evidence from published assays demonstrate that EO’s antioxidant activity might be due to the presence in their constitution of phenolic compounds, being the carvacrol, eugenol, geraniol, chavicol, menthol, linalool, citronellol and thymol those in larger concentrations [5]. Nevertheless, it is important to consider that, as pointed out by different assays, the bactericidal and antioxidant effects cannot be solely attributed to the predominant compounds. Equally significant is the interplay between these major compounds and those present in lower concentrations (Burt, 2004; Hyldgaard et al., 2012; Ojeda-Sana et al., 2013).”

As Myrtus communis L. is a plant that spontaneously grow in our country and available in north of Portugal, the authors considered interesting to evaluate the potential of its EO. We did not pretend to evaluate any compound isolated, but the global impact of this EO.

Reviewer´s comment:

Line 115: the authors mentioned that the aim of their studies was also “chemical composition by chromatographical techniques.” However I cannot see the data concerning this. Table with identified compounds together with retention indices as well as GC chromatogram should be provided even in the supplementary materials!!!

Our reply: The sentence was improved and a table with retention times was introduced in the supplementary materials.

Material and methods

Reviewer´s comment:

What king od standards were used for GC-MS?

Our reply: Authentic standards refers to GC-MS analysis of commercially available standards that have similar chemical properties as those which are identified in the EO samples (i.e., prenol). Quantification was performed via internal standard methodology, using compounds absent from the initial matrix. Calculations were performed taking into account the NL value – among other parameters – obtained for the signals of interest.

Reviewer´s comment:

Based on the description of GC-MS conditions I see that the analysis was performed in the standard GC columns not chiral. The question is how did the authors identify D-limonene, since under normal conditions there is no differences between D- and L-limonene (the same mass spectra and retention indices)? The same question concern (-)-trans-pinocarvyl acetate. Are you really sure that it was minus isomer, not plus???!!!! Did you check this?

Our reply: Literature (historically) indicates that D-Limonene is the predominant form in nature, as it is taught academically, whereas the laevorotatory isomer is less common. This paper does not focus on the potential antioxidant activity of any particular compound pertaining to the Myrtus communis L. EO, nor any compound isolation is performed toward that purpose. The assay infers that activity is most likely a result of synergy between different compounds, not a single compound, therefore unequivocal identification of the particular isomer, while necessary, is less relevant. Congruently, the chromatographical analysis was limited by the availability of better suited hardware (i.e., chiral columns). Compound identification was performed with a certain degree of automation, and name attribution (and extraction) has been done by name matching, being the data grouped in a singular tabular form, hence the existence of enantiomeric identifiers. For the sake of clarity, enantiomeric identifiers have been removed from the text and tables.

Reviewer´s comment:

How was the percentage of compounds in the EO calculated?

Our reply: The percentage of compounds in the EO was calculated considering the relative amount (% area) of the total identified compounds (83.23%). Please see table in supplementary materials.

Results

Reviewer´s comment:

The authors should provide the GC-MS data, even in the supplementary materials. I am doubtful about proper identification of EO components. Did the authors use the positive control in their studies?

Our reply: Myrtus communis L. is a well-known (and studied) plant and its chemical profile is easily found upon a brief survey on the literature. Several publications exist with this information, e.g., a sister publication within the MDPI group has a concisely summarized compendium of this plant chemical profiles from different countries: Usai et al., Molecules 2018, 23(10), 2502. Variations exist, as it is expected, that can be attributed, to genotype expression, climate conditions, among others, but the core compounds are systematic: 1,8-cineole, α-pinene, myrtenyl acetate, linalool, geranyl acetate and limonene. Analyte identification was performed by comparison of Kovats and linear retention indices, using NIST/EPA/NIH library (2011) and other libraries, and by comparison of true standards.

Discussion

Reviewer´s comment:

I am not sure whether the lower yield of essential oil is because of climatic conditions. It could be because of the drying temperature. The authors mentioned in the Material and methods part that plant material was dried in the temperature 40±0.5 oC. It is a bit too high for aromatic plants. It should be not more than 35 degrees. I understand that sometimes, because of the climatic conditions it is difficult to keep such a temperature, but it should be mentioned.

Our reply: In fact, plant material was dried at the temperature of 40 ºC. This was rectified in the text.

Reviewer 3 Report

In this study, the authors investigated to study antimicrobial effect of EOs extracted of Myrtus communis L. leaves at level of microbiota isolated from decaying fresh "Maronesa"meat determining their minimum inhibitory concentration and chemical composition by chromatographical techniques. The conclusions are sound and really summaries the novelty of the work. However, there is a lot of room for improvement and some issues should be addressed before publication.

<Materials and methods>

More information (name, city, country, etc) of materials, apparatus, and institutes should be indicated through the whole manuscripts.( ex. line 123, 125, 126, 132 etc)

Line 187 : How amount of EOs was added to the sample for each conditions?

<Results>

Line 251 : The differences of bacteria counts between Control and EO was not significant if the unit was 'cfu/g' which was only about 3cfu/g at maximum. So, please check the unit (cfu/g or log cfu/g) or the revise the sentences.

Figs : Please check the unit of bacteria counts (cfu/g or log cfu/g).

Line 262 : Even though the differences was the not significant, but you should explain why the difference occured in detail.

<References>

Please double-check the references with following the guideline of the journal

Author Response

The authors are grateful to the referee for the attentive and detailed remarks which helped to considerably improve the paper.

We hope the answers below and modifications introduced in the manuscript are clear and concise enough as required by the Reviewer.

All the revisions are marked in the text.

Reviewer´s comment:

1-      Keywords: I suggest to add the keyword active packaging

Our reply: Done as required.

Reviewer´s comment:

2- The illustrations are sharper, but the chosen colours and symbols still allow for improvements. I would also appreciate it if you would add some of the explanations given to me to the paper, so that the reader is also informed

Our reply: Amended as required.

Reviewer´s comment:

3-  All in all, I  miss the innovative character or the differentiation from other works. You should point out one of them and highlight what  needs to be investigated in further research or where there is potential for optimisation

Our reply: Amended as required. Sentences were introduced in conclusions.

Reviewer 4 Report

I am grateful for the opportunity to participate in the peer review of the aforementioned manuscript.

Based on my analysis, I believe the manuscript in question has a good chance of being accepted, albeit with some revisions.

Here are some remarks I would like authors to address:

- Line 23. Correct misspelled word "vaccum" for "vacuum" throughout all text.

- Line 28-29. Errors in strain designations. Did you mean ATCC BAA-769 for the LM (not just ATCC 679)? Also, ATCC 35150 should refer to O157:H7, as well as erratic NCTC 2900 which should stand as NCTC 12900.

- Line 73. Beware, Leuconostoc is a genus of gram-positive bacteria!

-  Line 96. Please, rephrase "notorious" since it indicates that myrtle leaves are famous for something bad which is not true.

- Line 114. Please, indicate that this DOP "Maronesa" is indeed a breed of mountain cattle.  It is not a plain meat, but a beef.

- Line 168. Please, update as remarked for Lines 28-29 and throughout the text.

- Line 213. Correct the expression. Plates were inoculated, not seeded.

- Line 219. Did you mean "Statistica" SW? Indicate sw manufacturer.

- Lines 227 -229 need to be rephrased and corrected. So, what are the 3 categories? Terpenes, terpenoids, and phenylpropanoids, right? Correct to "low quantities".

- Line 250 and 258 and 285 and 306. Correct ***P<0.00 to (I suppose) <0.001.

- Line 283. Why EO values in Table 3 for Entero, Pseudomonas, and fungi do not contain superscripts indicating statistical significance?

- Line 341 and 342. Error! Reference source not found.

- Line 407-408, please rephrase since it is confusing.

- Line 410. I disagree with this since you stated in Line 229 that phenylpropanoid content is the lowest (2.18%), so how come now it is "high percentage"?

- Throughout the text use the exact Degree symbol, not the underlined symbol. Also, do not mix EO and OE acronyms in the discussion.

Please, run the English proofreading in order to correct grammar mistakes in English and ambiguities pertaining to them.

Author Response

The authors are grateful to the referee for the attentive and detailed remarks which helped to considerably improve the paper.

We hope the answers below and modifications introduced in the manuscript are clear and concise enough as required by the Reviewer in order to enable the publication of the manuscript in Foods.

All the revisions are marked in the text.

Reviewer´s comment:

- Line 23. Correct misspelled word "vaccum" for "vacuum" throughout all text.

Our reply: Done as required.

Reviewer´s comment:

- Line 28-29. Errors in strain designations. Did you mean ATCC BAA-769 for the LM (not just ATCC 679)? Also, ATCC 35150 should refer to O157:H7, as well as erratic NCTC 2900 which should stand as NCTC 12900.

Our reply: The authors are grateful to the reviewer for the attentive remark. Amended as required.

Reviewer´s comment:

- Line 73. Beware, Leuconostoc is a genus of gram-positive bacteria!

Our reply: Amended as required.

Reviewer´s comment:

-  Line 96. Please, rephrase "notorious" since it indicates that myrtle leaves are famous for something bad which is not true.

Our reply: Amended as required.

Reviewer´s comment:

- Line 114. Please, indicate that this DOP "Maronesa" is indeed a breed of mountain cattle.  It is not a plain meat, but a beef.

Our reply: Amended as required.

Reviewer´s comment:

- Line 168. Please, update as remarked for Lines 28-29 and throughout the text.

Our reply: Amended as required.

Reviewer´s comment:

- Line 213. Correct the expression. Plates were inoculated, not seeded.

Our reply: Amended as required.

Reviewer´s comment:

- Line 219. Did you mean "Statistica" SW? Indicate sw manufacturer.

Our reply: Amended as required.

Reviewer´s comment:

- Lines 227 -229 need to be rephrased and corrected. So, what are the 3 categories? Terpenes, terpenoids, and phenylpropanoids, right? Correct to "low quantities".

Our reply: Yes. Amended as required.

Reviewer´s comment:

- Line 250 and 258 and 285 and 306. Correct ***P<0.00 to (I suppose) <0.001.

Our reply: Amended as required.

Reviewer´s comment:

- Line 283. Why EO values in Table 3 for Entero, Pseudomonas, and fungi do not contain superscripts indicating statistical significance?

Our reply: Because no differences were found for those microorganisms during time of storage. But, we agreed with the reviewer and added the superscripts.

Reviewer´s comment:

- Line 341 and 342. Error! Reference source not found.

Our reply: Doi was introduced in the reference.

Reviewer´s comment:

- Line 407-408, please rephrase since it is confusing.

Our reply: Amended as required.

Reviewer´s comment:

- Line 410. I disagree with this since you stated in Line 229 that phenylpropanoid content is the lowest (2.18%), so how come now it is "high percentage"?

Our reply: The sentence was corrected as required.

Reviewer´s comment:

- Throughout the text use the exact Degree symbol, not the underlined symbol. Also, do not mix EO and OE acronyms in the discussion.

Our reply: Amended as required.

Comments on the Quality of English Language

Please, run the English proofreading in order to correct grammar mistakes in English and ambiguities pertaining to them.

Our reply: The English language was revised over the text.

Round 2

Reviewer 2 Report

The manuscript looks better after reviewing process. However, before publication it needs some minor correction.

Since the authors used relative percentages which is very semiquantitative method, please display your results accurate to one decimal place.

There is no reference in the text to the table included in the supplementary materials

Compound names should be corrected (in the text and table):

α-terpinyl acetate insteat of α-terpineol acetate
β-caryophyllene instead of caryophyllene (it is because humulene is also caryophyllene but α)

lines 300-301; the same compound was mentioned two times: 'α-terpineol acetate (2.2%), α-terpineol acetate (2.22%)' . SO, please correct and use the proper name α-terpinyl acetate

Author Response

DETAILED RESPONSE TO REVIEWER 2

GENERAL COMMENTS TO THE AUTHOR

Reviewer´s comment:

The manuscript looks better after reviewing process. However, before publication it needs some minor correction.

Our reply: The are grateful to the referee for the attentive and detailed remarks which helped to considerably improve the paper.

We hope the modifications introduced in the manuscript are clear and concise enough as required by the Reviewer in order to enable the publication of the manuscript in Foods.

All the revisions are marked in the text.

Reviewer´s comment:

Since the authors used relative percentages which is very semiquantitative method, please display your results accurate to one decimal place.

Our reply: Amended as request.

Reviewer´s comment:

There is no reference in the text to the table included in the supplementary materials

Our reply: Amended as request.

Reviewer´s comments:

Compound names should be corrected (in the text and table):

α-terpinyl acetate insteat of α-terpineol acetate
β-caryophyllene instead of caryophyllene (it is because humulene is also caryophyllene but α)

lines 300-301; the same compound was mentioned two times: 'α-terpineol acetate (2.2%), α-terpineol acetate (2.22%)' . SO, please correct and use the proper name α-terpinyl acetate

Our reply: Amended as request.
